# A Cloud Detection Algorithm with Reduction of Sunlight Interference in Ground-Based Sky Images

**Xin Li, Zhiying Lu \*, Qingxia Zhou and Zhengyang Xu**

School of Electrical and Information Engineering, Tianjin University, Tianjin 300072, China;
xinlitu@tju.edu.cn (X.L.); zhouqx@tju.edu.cn (Q.Z.); xuzhengyang@tju.edu.cn (Z.X.)

**\*** Correspondence: luzy@tju.edu.cn

**Abstract:** Cloud detection for ground-based sky images has attracted much attention in cloud-related fields. In this paper, we proposed a cloud detection algorithm that reduced the sunlight interference in the image. The solar location method was introduced to track the sun in the image used for feature calculation, which was suitable for the case where the camera could not be calibrated. Following this, the adjustable red green difference (ARGD) feature using red and green channels was proposed. The red weight in the feature was determined by the layering region division, which classified the degree of sunlight interference in the image, and the sky state, which discriminated whether there was sunlight interference in the image. Finally, a fixed zero threshold was applied to feature images in order to obtain the cloud detection results. Experimental results showed that the proposed algorithm performs better than the other algorithms and can effectively reduce the sunlight interference.

**Keywords:** clouds detection; sunlight interference; solar location; ground-based images; image processing

## 1. Introduction

Clouds influencing the Earth's radiation budget and the energy balance continue to contribute to the largest uncertainty for climate models and climate predictions [1]. The analysis of cloud behaviors based on total-sky imagery (TSI) cloud detection is of great significance for climate change research. Reference [2] pointed out that the diurnal to seasonal variation research of the subtropical low clouds over the northeast Atlantic has many important implications. The uncertainty of solar generation is related to the movement of clouds that leads to ramp events [3]. Cloud research and characterization are based on cloud detection, but sunlight interference is a major challenge improving the accuracy of cloud detection.

The observation of clouds and clear skies has long relied on the use of different optical characteristics and their combinations. Theoretically, many optical features originate from the fact that the short-wave scattering makes clear skies appear blue, meaning that the blue component is greater than the red component, while the uniform scattering of visible-light wavelengths makes the clouds appear white, indicating the similar amounts of red and blue components [4]. Given that the red, green, and blue (RGB) color space is closely related to the computer storage of images and the theoretical origin of color, RGB has gradually become one of the most widely used characteristics.

The red blue ratio (RBR) was implemented for the first time in the cloud detection algorithm of whole sky imagers, which has been widely used for decades [5,6]. Reference [7] originally proposed the red-blue difference (RBD) that directly used the theoretical origin to create a difference between the red and the blue channels of the images. The normalized B/R ratio (NRBR) enhances the image contrast and exhibits robustness to noise in application to cloud detection [8,9]. A detailed comparison analysis was performed for the above features. Reference [10] pointed out that RBR takes into account

the chrominance and luminosity, while RBD cannot be considered to be a function of luminosity. During image acquisition, the down-sampling caused by the JPEG format reduces the resolution of chrominance, but the luminosity is not affected, resulting in the higher resolution of RBR. By the equation transformation, NRBR can be shown to be a monotonic decrease of RBR. Thus, these two features have different advantages and are widely used as classic features in cloud detection.

In addition to RGB, other color spaces are also used for cloud detection. An algorithm that distinguished clouds with gray distribution from clear skies only used the saturation channel from the hue, saturation, and value (HSV) color space [11]. This feature is not only suitable for high latitude regions, but also shows excellent performance for cloud detection in general regions. To fully extract the image information, a color model based on the RBR and channels of three color spaces, including RGB, HSV, and luminance, blue-difference chroma, and red-difference chroma (YCbCr), was applied to cloud detection using multiple classifiers [12]. Moreover, the application of multicolor spaces is a popular trend in image recognition. Combinations of multiple methods have also been developed to identify clouds. The Euclidean geometric distance, used to measure the difference between the RGB vector and the space diagonal, was combined with the Bayesian method, used for classification in order to identify cloud pixels [13].

Sunlight with non-uniform brightness increases misclassification and hinders the optimization of algorithms. To reduce the negative effects of sunlight, [14] proposed that half of the sun circle area cloudiness should be reduced when the cloudiness in the sun circle area was overestimated. This method corrects erroneous sky coverage, but a flexible approach is more suitable for the changing sky. This led to the development of the differencing method, in which a clear sky library is established and then a feature image is compared to the feature image of the corresponding clear sky from the library [10]. Better results were obtained using this method compared to those obtained by the traditional methods, particularly in the circumsolar and near-horizon regions in the case of the sunlight affecting the image [15]. Since the clear sky images are affected by sunlight, the effect in the original image is eliminated after the contrast. However, if images do not have sunlight interference, detection errors will be obtained. Therefore, [16] attempted to extract cloud pixels using the single threshold strategy in the sun-invisible cases or the differencing method in the sun-visible cases, obtaining better detection results.

Statistical analysis methods have been used to obtain good detection results in the circumsolar and near-horizon region without creating a clear sky library. Reference [17] divided the sky into three categories of cloudless, partially cloudy, and overcast, and constructed a separate model for each category. Channels and channel combinations from RGB and HSV color spaces were studied in detail, and then the criteria of these characteristics were incorporated into the model in order to determine the distribution of the clouds. Analogously, a multicolor criterion was proposed to estimate cloud coverage, obtaining improved results for broken and overcast cloudiness [18].

The image is transformed by a certain feature, and then one or several thresholds are set to segment the feature image into binary images. In an early thresholding method, the fixed threshold was obtained based on experience or image statistics and the segmentation employed a logical operation with a very small computational cost to distinguish the cloud pixels. Another method adopted the adaptive threshold suitable for the case where the image histogram exhibits double peaks, that is, both cloud pixels and clear sky pixels exist and differ strongly [19–21]. Adaptive thresholding showed better performance than fixed thresholding for images with cumuliform and cirriform clouds. Reference [22] proposed a hybrid thresholding algorithm (HYTA) that combined the advantages of the fixed thresholding and the adaptive thresholding.

The paper proposed a cloud detection algorithm that reduced the sunlight interference in the image. Section 2 describes the image acquisition equipment, the image set, and the preprocessing. Section 3 describes the cloud detection algorithm in detail. Section 4 presents the results and discussion, and Section 5 provides concluding remarks.

## 2. Data

The images used in this study were collected from the Total Sky Imagery model 440 (TSI-440) instrument manufactured by Yankee Environmental Systems (Turners Falls, MA, USA). The charged-couple device (CCD) camera, embedded at the top of the TSI, overlooked the hemispherical mirror to capture the reflected sky conditions. The shadow band on the mirror prevented strong solar radiation from damaging the imaging system. The parameter settings for sampling and image processing were configured in the TSI software, which was installed on the user's computer. The size of the sky images used was 640 × 480, and the image format was JPEG. Sky images had high temporal and spatial resolution, providing more accurate regional sky conditions. The images used in this work were provided by the Atmospheric Radiation Measurement (ARM) Climate Research Facility, and the image acquisition device was located at 39.0916° N, 28.0257° W: Eastern North Atlantic (ENA) Graciosa Island, Azores, Portugal.

The images were selected from 2015 to 2016 to cover the entire year in order to test the universality of the algorithms for the four seasons and different times. According to the standard of two images per day, 730 images were randomly selected by generating uniform random numbers corresponding to the running time of the machine. Figure 1 shows the time distribution of the selected images. The horizontal axis indicates the date of image generation, while the vertical axis represents the time of image acquisition. The blue curve consists of many blue circles representing the start time of the device, and the red curve consists of red circles indicating the end of the device operation. The samples are shown by green circles demonstrating the uniform distribution of the samples. The scattered and fixedly positioned 18 image patches in each image were selected. These patches were manually labeled as clouds or clear skies by specialists, which were regarded as the ground truth to evaluate the detection performance.

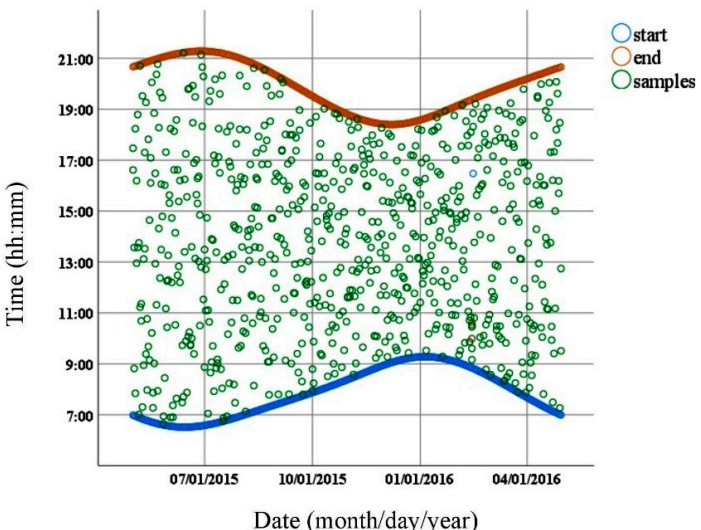

**Figure 1.** Time distribution of selected images.

The sky images were preprocessed. An image consists of the sky region and the rest of the image. The sky region is the region of interest of the image, while the rest of the image is redundant. The rest of the image, including the surroundings, the low altitude angle area in the sky, and occlusions caused by the device itself, were removed for more accurate and rapid cloud detection (see Figure 2a). First, the region located in the available field of view was intercepted, eliminating the first two redundancies. Then, the occlusions were positioned and repaired by algorithms, which was done by other members of the team. After the preprocessing, only the sky region used for cloud detection remained, as shown in Figure 2b.

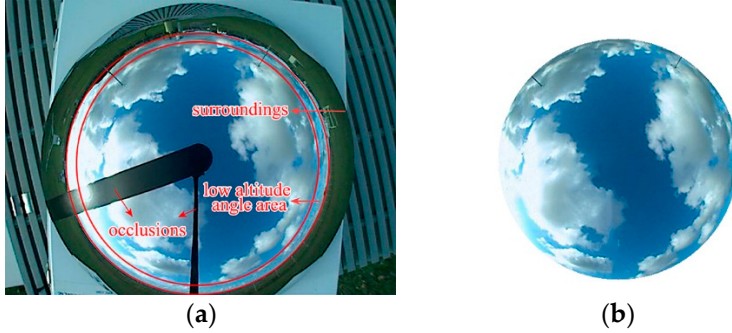

<div style="text-align:center">(**a**) (**b**)</div>

**Figure 2.** Redundancy in the image and the pre-processing result: (**a**) Redundancy in the image; (**b**) the preprocessing result.

## 3. Methodology

The cloud detection algorithm is illustrated in detail in Figure 3. First, the solar location method is described to track the sun in the image used for feature calculation, which was suitable for the case where the images were taken from the platform and camera calibration using the OcamCalib software toolbox was not possible [23]. Then, the layering region division and the sky state used to determine the red weight in the feature are presented. Next, the ARGD feature used to reduce sunlight interference is introduced. Finally, the pixel identification, based on a zero threshold, is described. To illustrate the performance of the proposed feature, ARGD was compared with classical features such as NRBR and RBR. We noted that the three features use RGB channels. Fixed thresholding and adaptive thresholding are popular methods of images segmentation, while zero threshold is applied to ARGD feature images.

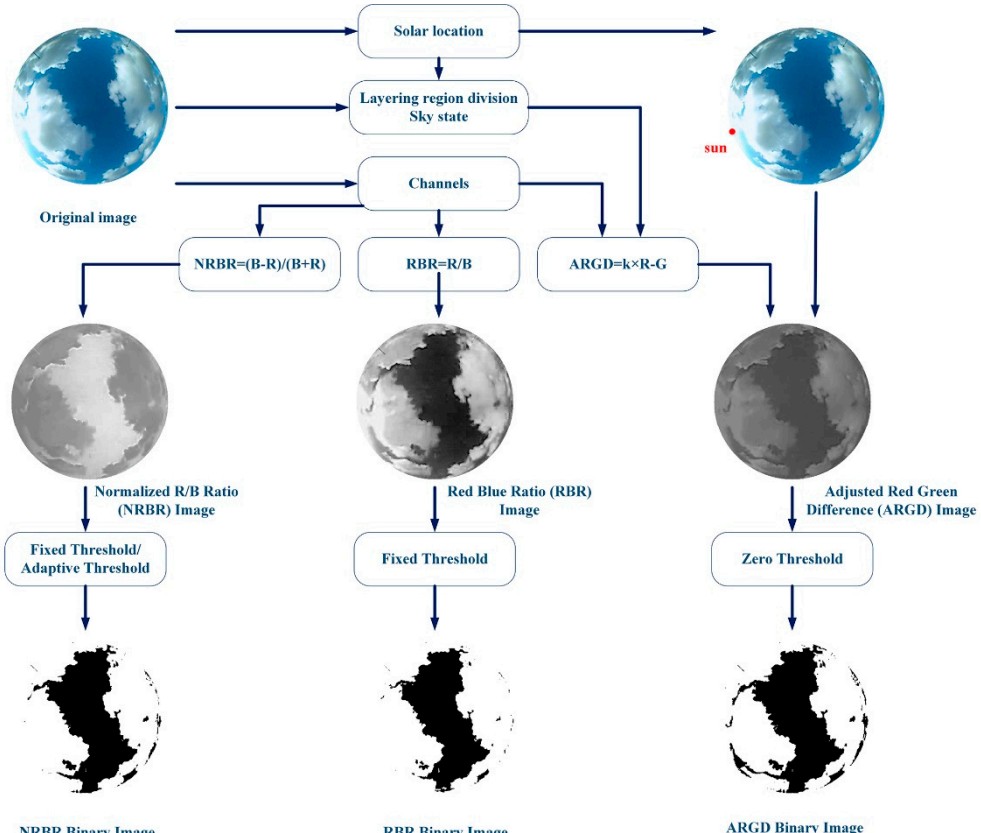

**Figure 3.** System framework and comparison of features (adjustable red green difference (ARGD), normalized red/blue ratio (NRBR), and red blue ratio (RBR)).

### 3.1. Solar Location

Limited to the conditions in which the camera cannot be calibrated, a solar location method was proposed that did not use any calibration object. First, some images in which the sun can be clearly identified were selected. Then, the coordinates of the sun in the horizontal coordinate system, the solar zenith angle (SZA), and the solar azimuth angle (SAA) were calculated from the astronomical formula. Next, the coordinates of the sun in the pixel coordinate system, namely the row and column of the pixel, were determined. Finally, the coordinate transformation from the horizontal coordinate to the pixel coordinate was established. In this way, the position of the sun in the image was determined by performing the coordinate transformation on the horizontal coordinates obtained by real-time calculation.

In this work, a total of 1323 images were selected in which the sun can be clearly identified. The images were randomly selected for universality and contained complete information, such as date, time, latitude, and longitude.

The celestial coordinate system divides the sky into two hemispheres: The upper hemisphere, where the objects above the horizon are visible, and the lower hemisphere, where the objects below the horizon cannot be seen. The pole of the upper hemisphere is called the zenith. A celestial coordinate system is a horizontal coordinate system that uses the observer's local horizon as the fundamental plane. The spatial position of an object is expressed in terms of horizontal coordinates including SZA and SAA. Figure 4 visually shows the position of the sun in the horizontal coordinate system, where $\alpha$ is the SZA representing the angle between the zenith and the center of the solar disk, and $\beta$ is the SAA meaning the angle of relative direction of the sun along the local horizon. The units of SZA and SAA, expressed as floating-point numbers retaining 15 significant digits, are degrees. The SZA with the range of (0°, 180°) was measured toward the ground with the zenith as the reference, and the SAA with the range of (0°, 360°) was increased clockwise from the south. The SZA was calculated from the date (year, month, day), time (hour, minute, and second), and the latitude and longitude:

$$\alpha = \arccos(\sin\delta\sin\varphi + \cos\delta\cos\varphi\cos\tau), \tag{1}$$

where $\delta$ is the sun declination angle calculated based on the date, $\phi$ is the local geographic latitude, and $\tau$ is the solar hour angle computed based on the date, time and longitude. The SAA is calculated as follows:

$$\beta = \arccos(\frac{\sin\alpha\sin\varphi - \sin\delta}{\cos\alpha\cos\varphi}). \tag{2}$$

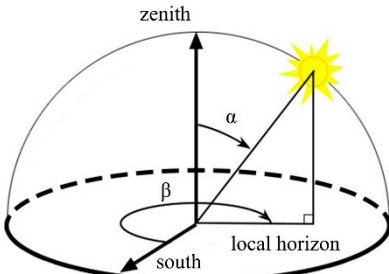

**Figure 4.** Spatial position of the sun.

Thus, the horizontal coordinate of the sun can be derived from the date, time, longitude, and latitude in real time.

After obtaining the horizontal coordinate of the sun, the pixel coordinate must also be determined. The pixel coordinates are the specific locations of pixels in the image, namely the row and column in pixels. Both row and column are integers in the range of (0, 255). The sun in the image was manually signed and the pixel coordinates of the sun were extracted.

To illustrate the relationship between the horizontal coordinates and the pixel coordinates, three-dimensional scatter plots were drawn. Here, the SZA and SAA of the horizontal coordinates were taken as the x-axis and the y-axis, respectively, and the row and column of the pixel coordinates were sequentially taken as the z-axis. As a result, two scatter plots, a row-scatter plot, and a column-scatter plot were obtained. Figure 5 shows that two scatter plots exhibited the surface distribution taking into account the errors of the manual annotation.

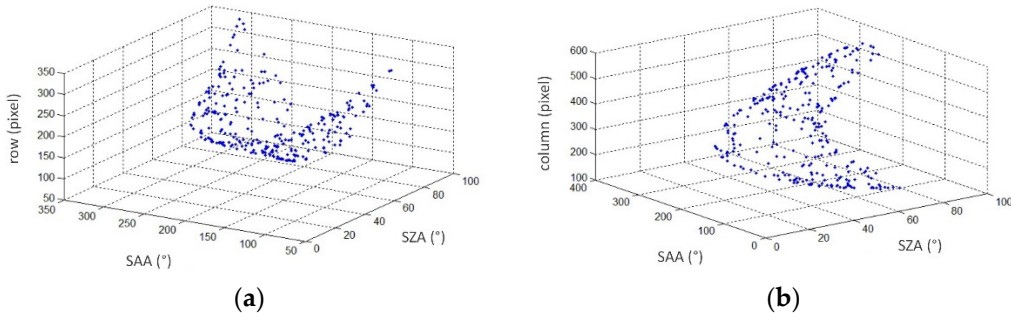

**Figure 5.** Row-scatter and the column-scatter plots: (**a**) The row-scatter plot; (**b**) the column-scatter plot.

The appropriate fitting method based on scatter plots was performed to obtain the coordinate transformation formula. In this work, polynomial surface fitting was carried out. We attempted to take into account the various combinations of SZA and SAA from the first to fifth order. The SZA is denoted by $\mathbf{Z} = [1, \alpha, \alpha^2, \alpha^3, \alpha^4, \alpha^5]$. The SAA is denoted by $\mathbf{A} = [1, \beta, \beta^2, \beta^3, \beta^4, \beta^5]^{\mathrm{T}}$. The coordinate transformation formula were determined as follows:

$$c = \mathbf{Z} \begin{bmatrix} p_{00} & p_{01} & p_{02} & p_{03} & p_{04} & p_{05} \\ p_{10} & p_{11} & p_{12} & p_{13} & p_{14} & 0 \\ p_{20} & p_{21} & p_{22} & p_{23} & 0 & 0 \\ p_{30} & p_{31} & p_{32} & 0 & 0 & 0 \\ p_{40} & p_{41} & 0 & 0 & 0 & 0 \\ p_{50} & 0 & 0 & 0 & 0 & 0 \end{bmatrix} \mathbf{A}. \tag{3}$$

where $c$ is the row or column of the predicted pixel coordinate and $p_{ij}$ are coefficients. The coefficients were determined by continuous optimization (see Section 4.1). The obtained results show that the row-scatter plot had the best performance on the second-order SZA and the fourth-order SAA, while the column-scatter plot showed the best performance from the second-order SZA and the third-order SAA.

In this way, the pixel coordinates of the sun in the images were determined using the SZA and SAA as input into the transformation formula.

### 3.2. Cloud Detection Feature

Cloud detection often fails for images that suffer from interference by sunlight. The saturation enhancement caused by strong light gives rise to inaccurate detection of many excellent algorithms. In this work, a cloud detection feature was proposed to reduce the influence of sunlight.

### 3.2.1. Layering Region Division

The interference of sunlight in the image was closely related to the solar position and was not uniformly distributed over the entire image. The circumsolar regions suffered the most serious interference, while the interference was drastically weakened as the distance from the sun.

The layering region division was applied to classify the degree of sunlight interference in the image. Three sun-centered circles with different radii of 90, 120, and 150 pixels, respectively, were set, and the image was divided into four layers (see Figure 6). The choice of radius was based on the visual observation. The first layer was a circular area containing the sun and was most affected by the sunlight.

Generally, this area had the greatest color change in an image and was easily saturated. The second layer, appearing as a ring, was close to the sun, but was less affected than the first layer. Hence, the color change was somewhat stable. In the third layer, which still appears as a ring, the influence of sunlight was weakened again, together with the color change. In the remaining area, set as the fourth layer, the color change was stabilized and sunlight interference is close to zero.

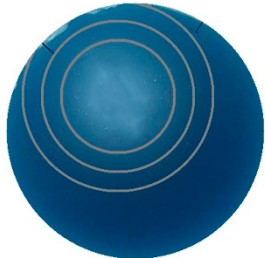

**Figure 6.** Layering region division.

### 3.2.2. Sky State

Both clouds and clear skies have different characteristics in different sky states. If the image displays interference caused by sunlight, the sky characteristics are different between the layers and the detection criteria are also different. If the image does not show sunlight interference, the differences between the sky characteristics of the different layers are small and a unified detection standard can be used. To distinguish the sky states that display sunlight interference and no sunlight interference, two parameters, the solar intensity (SI) and saturation difference (SD), were set.

The average intensity of the pixel block taken from the center of the solar disk was calculated as the SI:

$$\text{SI} = \frac{\sum\limits_{i=1}^{n} I_i}{n},\tag{4}$$

where $n$ is the number of pixels in the pixel block, $i$ is the sequence number of the pixel, and $I_i$ represents the intensity of the $i$-th pixel. The dimensionless parameter SI is an integer value in the range of (0, 255). The pixel intensity reaches the maximum value because the pixels of the sun disk region tend to be saturated in sunny conditions, while the intensity is significantly lower when clouds block the sun. The SI indicates whether the sun appears in the image. The appearance of the sun produces a certain degree of interference with the image. The use of SI as the first parameter for evaluating the sky state is enabled by its simple calculation and excludes most of the images, significantly reducing the computational cost of the algorithm.

To determine whether the image has interference caused by sunlight, the SD was calculated by subtracting the average of the saturation of the first layer from that of the entire image as expressed by:

$$\text{SD} = \frac{\sum\limits_{i=1}^{n_w} S_i}{n_w} - \frac{\sum\limits_{j=1}^{n_1} S_j}{n_1}.\tag{5}$$

where $n_w$ is the number of pixels in the entire image, $n_1$ is the number of pixels in the first layer, and $S_i$ is the saturation of the $i$-th pixel. The dimensionless parameter SD is a floating-point value in the range of (0, 1). The saturation distribution of the clear sky with sunlight interference gradually increased from the first layer to the first four layers. However, the saturation distribution was uniform when the image had no interference. The saturation of the first layer determines whether there are clear skies or clouds with color changes affected by sunlight and indicates the state of the circumsolar regions. Analogously, the saturation of the entire image indicates the state of the entire sky. The SD indicates whether the sky states of the first layer and the entire image are consistent. If the parameter value

is small, it can be determined that the saturation of the sky is evenly distributed, and the sky has no interference by sunlight. Otherwise, the sky is affected by sunlight.

The thresholds for distinguishing the sky state were obtained as SI of 180 and SD of 0.1. The threshold acquisition is detailed in Section 4.2. The decision model is given: If SI is less than 180 and SD is less than 0.1, the image has no interference by sunlight. Otherwise, the image has sunlight interference.

### 3.2.3. ARGD Feature

The ARGD feature with strong adaptability was proposed using the red and green channels according to the following definition:

$$ARGD = k \times R - G \tag{6}$$

where $k$ is the weight of the red channel. If the sky state indicated that the image had no interference of sunlight, the weights for layers 1–4 were set to 1.3, 1.4, 1.5, and 1.7, respectively. Otherwise, the weight in each layer was set to the constant value of 1.7. The weights were obtained by analysis of linear histograms.

Both the layering region division and the sky state are considered to determine the weight. If the sky state indicates the presence of sunlight interference, the difference between the red and green channels of the clouds is gradually increased from the first layer to the fourth layer, and that of clear skies is reduced. Since each layer has different characteristics, it is necessary to set different weights. If the sky state does not indicate the presence of sunlight interference, the weight set to constant is reasonable. Through the adjustment to the weight, the feature can adapt to the characteristics of different layers and meet the different detection requirements.

The study of channels for clouds and clear skies was carried out (see Section 4.3). The analysis showed that the green channel had the stable centralization and the blue channel was easily saturated. From the perspective of standardization, ARGD uses red and green channels.

It is important to note that the zero threshold satisfies the requirement of segmentation. When the feature values are positive, pixels are identified as clouds, and if the values are negative, the pixels are determined to be clear skies.

The practical example of cloud detection is shown in Figure 7. The sky image is shown in Figure 7a. The sky state determined the sunlight interference by calculating SI and SD, and the weight was determined accordingly. Figure 7b shows the detection result when the weight has a constant value of 1.3, and the result in the first layer was adopted. Figure 7c,d show the detection results when weights were set to constant values of 1.4 and 1.5. The results of the ring regions in the second and third layers were adopted. Figure 7e shows the detection result when the weight was set to a constant value of 1.7, and the results of the remaining areas were adopted. Figure 7f shows the final detection result of the ARGD feature under zero threshold. As observed from the results of the processing, the amount of identified clouds increased with increasing weight. The increased amount of clouds was incorrect in the circumsolar region with sunlight interference, but the clouds with the correct identification in this area were added, proving that this feature is reasonable.

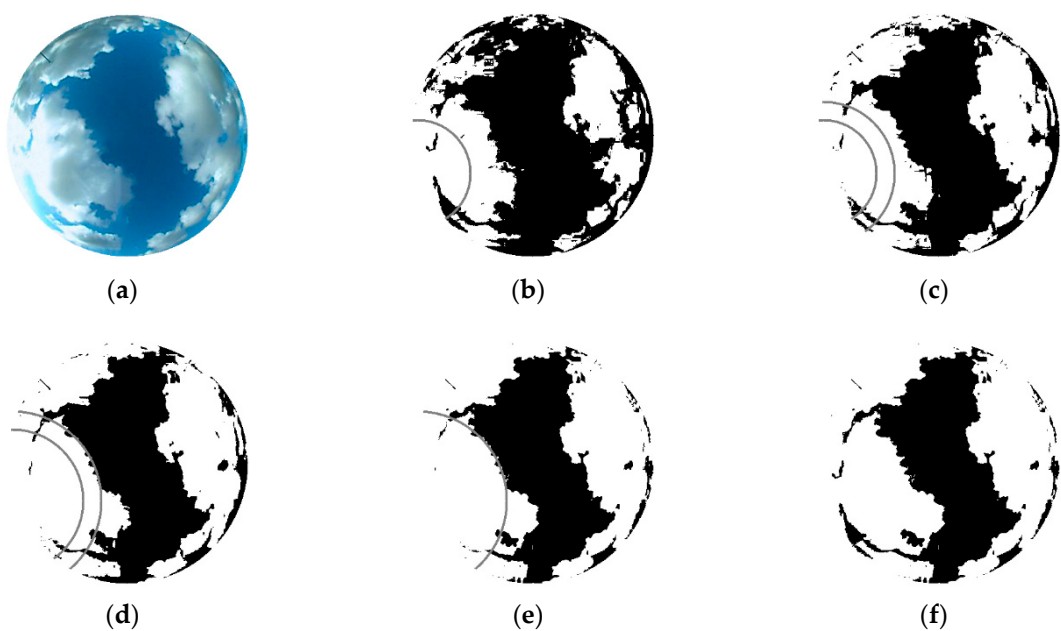

**Figure 7.** ARGD processing: (**a**) Original image; (**b**) ARGD binary image with *k* =1.3; (**c**) ARGD binary image with *k* = 1.4; (**d**) ARGD binary image with *k* = 1.5; (**e**) ARGD binary image with *k* = 1.7; (**f**) ARGD binary image.

## 4. Results and Discussion

### 4.1. Coordinate Transformation Formula

In the solar location part of the algorithm (see Section 3.1), the coordinate transformation from the horizontal coordinates to the pixel coordinates was established. In this way, the time and location information were used to calculate the SZA and SAA, and then the coordinate transformation was used to obtain the solar position in the images.

The coordinate transformation was constructed by polynomial surface fitting. The specific polynomials were determined according to the row and column scatter plot in order to take into account the various combinations of SZA and SAA from the first to fifth order. Two evaluation indicators, namely root mean square error (RMSE) and adjusted coefficient of determination (Adjusted R-square, $R^2$), were selected to find the best polynomials and were defined as:

$$\text{RMSE} = \sqrt{\frac{\sum\limits_{i=1}^{n}(y_i - \hat{y}_i)^2}{n}}, \tag{7}$$

$$\overline{R}^2 = 1 - \frac{\sum\limits_{i=1}^{n}(y_i - \hat{y}_i)^2}{\sum\limits_{i=1}^{n}(y_i - \overline{y})^2}\left(\frac{n-1}{n-p-1}\right). \tag{8}$$

where $y_i$ is the actual value of the *i*-th sample, $\hat{y}_i$ is the predicted value of the *i*-th sample, $\overline{y}$ is the average of the actual values, $p$ is the number of explanatory variables, and $n$ is the sample size. The RMSE value indicates the dispersion of the sample and the RMSE value close to zero indicates that the model fit well. The Adjusted R-square value ranges from 0 to 1, and a value closer to 1 indicates a better fit of the model for the samples. In this work, the RMSE and the Adjusted R-square described the fitting error of pixel coordinates row and column. The units of row and column are pixels.

The results of the row coordinate transformation for various combinations of orders are listed in Table 1. It can be observed that the row converting coordinate had the best performance in the

fourth order SZA and the fifth order SAA (see the bold numbers in Table 1). Considering the various cases of SZA to RMSE, the results of the first order SAA were the worst for all orders and the gap was large, which means that the data were not distributed in the first order. The fifth-order SAA were the best, indicating that the data were suitable for the fifth-order distribution of SAA. The Adjusted R-square was extremely inferior to first-order SZA and first-order SAA, which may be attributed to the difference between the three-dimensional surface and the three-dimensional plane. When SAA was not first order, the Adjusted R-square reached values above 0.9. Although the result of the fourth order and fifth order was not unique, it still had the best value.

**Table 1.** Results of the optimal row coordinate transformation.

| SAA Order SZA Order | First | Second | Third | Fourth | Fifth |
|---|---|---|---|---|---|
| **RMSE (Pixel)** | | | | | |
| first | 57.41 | 13.21 | 6.88 | 2.983 | 2.463 |
| second | 54.26 | 8.134 | 5.411 | 1.922 | 1.881 |
| third | 53.6 | 5.488 | 5.419 | 1.873 | 1.834 |
| fourth | 52.39 | 4.317 | 4.19 | 1.875 | **1.791** |
| fifth | 51.7 | 4.216 | 4.038 | 1.797 | 1.794 |
| **Adjusted R-Square (Pixel)** | | | | | |
| first | 0 | 0.9468 | 0.9856 | 0.9973 | 0.9981 |
| second | 0.1016 | 0.9798 | 0.9911 | 0.9989 | 0.9989 |
| third | 0.1235 | 0.9908 | 0.991 | 0.9989 | 0.999 |
| fourth | 0.1627 | 0.9943 | 0.9946 | 0.9989 | **0.999** |
| fifth | 0.9896 | 0.9946 | 0.995 | 0.999 | 0.999 |

Table 2 shows the results of the column coordinate transformation, and comprehensive analysis shows that the combination of the fifth-order SZA and the fifth-order SAA was the best (see the bold numbers in Table 2). The RMSE had a wide range, and the best result can be understood as corresponding to the sun's position fitted to be, on average, 1.505 pixels away from the marked position. The Adjusted R-square of any combination reached 0.9 or higher. Therefore, an appropriate column coordinate transformation was fitted.

The method for locating the sun provides a solution in the case where the information is insufficient and where the equipment cannot be tested experimentally. This situation is quite common. For example, when the data are obtained from a public platform, a research institution, a university, or an enterprise, the software interface of the equipment cannot be examined, and there is also no way to use the OcamCalib toolbox to calibrate the camera. Our method can locate the sun in the image using only the image, time, and device location.

**Table 2.** Results of the optimal column coordinate transformation.

| SZA Order ＼ SAA Order | First | Second | Third | Fourth | Fifth |
|---|---|---|---|---|---|
| **RMSE (Pixel)** | | | | | |
| first | 27.72 | 21.13 | 5.326 | 2.753 | 2.345 |
| second | 21.42 | 21.17 | 2.9 | 1.947 | 1.537 |
| third | 19.56 | 17.32 | 2.873 | 1.92 | 1.53 |
| fourth | 17.78 | 16.38 | 2.013 | 1.868 | 1.509 |
| fifth | 17.44 | 16.41 | 1.742 | 1.537 | **1.505** |
| **Adjusted R-square (Pixel)** | | | | | |
| first | 0.9491 | 0.9704 | 0.9981 | 0.9995 | 0.9996 |
| second | 0.9696 | 0.9703 | 0.9994 | 0.9997 | 0.9998 |
| third | 0.9747 | 0.9801 | 0.9995 | 0.9998 | 0.9998 |
| fourth | 0.9791 | 0.9822 | 0.9997 | 0.9998 | 0.9998 |
| fifth | 0.9799 | 0.9822 | 0.9998 | 0.9998 | **0.9998** |

*4.2. Parameters for Sky State*

Depending on the state of the clouds and sun, the images were divided into four categories, as shown in Figure 8. The first category of overcast sky is shown in Figure 8a where the sun was not observed and the sky was generally covered by clouds. The second type is cloudy without sun, shown in Figure 8b, where there were both clouds and clear skies, but the sun was blocked by the clouds. Figure 8c shows an image with clouds and clear skies that was placed in the cloudy with sun category because of the visible sun. The final category is that of clear sky, as shown in Figure 8d, where the sun was extremely clear and few clouds appeared

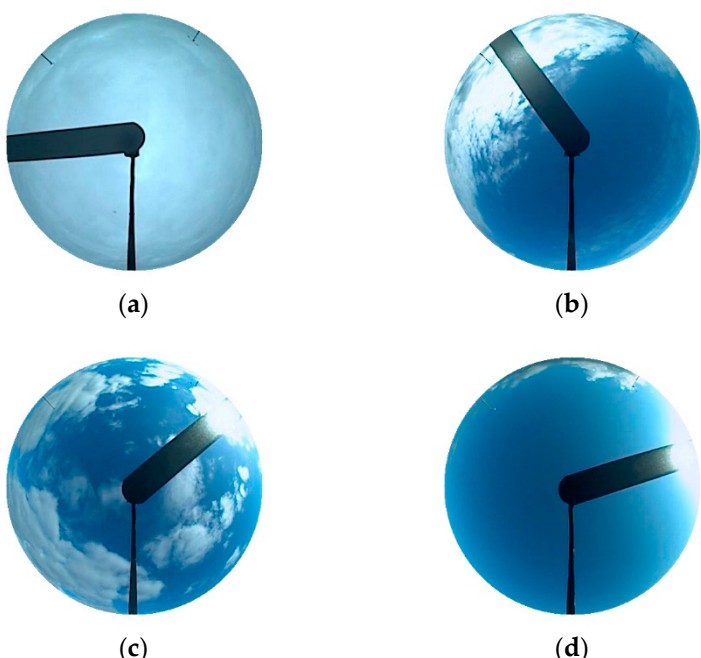

(a)　　　　　　　　　　　　　(b)

(c)　　　　　　　　　　　　　(d)

**Figure 8.** Four categories of images: (**a**) Overcast sky; (**b**) cloudy sky without sun; (**c**) cloudy sky with sun; (**d**) clear sky.

To determine the sky state, two parameters, namely SI and SD, were proposed (see Section 3.2.2). The SI was obtained by calculating the average intensity of the pixel block at the center of the solar disk. The statistical results are listed in the SI column of Table 3. The SIs of the cloudy skies without sun and overcast skies images were lower than those of the other image categories and their values were quite different. Moreover, the images in the overcast sky category had a lower SI value than the images for cloudy skies without sun. To identify whether the image had sunlight interference, the overcast sky without interference was distinguished from other categories and the threshold was set to 180.

**Table 3.** Statistical results of SI, S of different layers, and SD for four categories of images.

| Sky | SI | S1 | S2 | S3 | S4 | SD |
|---|---|---|---|---|---|---|
| Overcast | (28.81,176.92) | (0.19,0.50) | (0.21,0.48) | (0.23,0.46) | (0.26,0.42) | (0,0.08) |
| Cloudy without Sun | (62.03,197.44) | (0.15,0.32) | (0.19,0.36) | (0.22,0.41) | (0.35,0.59) | (0.11,0.36) |
| Cloudy with Sun | (244.85,252) | (0.14,0.36) | (0.21,0.42) | (0.27,0.48) | (0.40,0.67) | (0.17,0.44) |
| Clear | (247,252) | (0.21,0.58) | (0.29,0.64) | (0.35,0.68) | (0.55,0.77) | (0.19,0.37) |

Given the layering region division, the average saturation (S) of the first layer to the first four layers was calculated, and the ranges are listed in the S1–S4 columns in Table 3. The saturation was mostly either increasing or decreasing from the first layer to the first-fourth layer for any category, but there were some images in which the saturations of the first-second and first-third layers changed in the vicinity of the saturations of the first and first-fourth layers. Fifty images were taken randomly. For 90% of the images, the saturations were either increasing or decreasing, and for 10%, turbulent changes were observed. The first layer and the first four layers fully represented the changes in the saturation. For the first layer, the saturations of the images in the cloudy skies and clear skies categories had the large range of values due to the sunlight and clouds. The saturation of the clear sky was generally relatively large, but the saturation in the morning and evening led to the low value. For the first-fourth layer, the saturation increased sequentially from overcast to clear skies. The difference between the first-fourth layers and the first layer indicates whether the sky state of the first layer is consistent with that of the entire image and distinguishes the overcast skies from the other categories (see the SD column in Table 3). The SD of overcast skies was less than 0.08 and the threshold was set to 0.1.

*4.3. Analysis of Channels and Features*

To distinguish clouds and clear skies, the channels were analyzed. The RGB values are separately extracted along the red stripe in Figure 8a, and the distribution of the values representing the clouds and clear skies is presented in Figure 8b. The clear sky pixels were distinctly observed at 0–75 and 180–240, and cloud pixels were distributed in the 75–180 range. Bright cloud pixels were mainly distributed at 75–130, and dark cloud pixels were distributed over 130–180. It can be clearly observed that the two kind of clouds differed in that the channel value of the bright cloud pixels was larger than that of the dark cloud pixels. It is important to mention that the blue component of the bright cloud pixels was easily saturated because this component was limited to the range of 0–255. Of course, it is also possible that all three channels were saturated. It can be observed that the blue component was greater than the green component and the red component and the values of the cloud pixels were higher than those of clear sky pixels in each color component.

An unexpected but vital clue was contained in the rule of stable centralization of the green component, which was a ubiquitous phenomenon in the images used in our work. For further analysis, the chromaticity coordinates were presented as:

$$r = \frac{R}{R+G+B}, \; g = \frac{G}{R+G+B}, \; b = \frac{B}{R+G+B}. \tag{9}$$

This expression reflects that $r$ and $b$ were presented in axisymmetric form with $g$ as the axis, as shown in Figure 8c. The $g$ was stable at the value of 0.33, verifying the rule. If two of the three channel values are known, the remaining channel value can be approximated. This implies that the complete pixel state can be expressed with only two channels. The $r$ and $b$ values of the clear sky pixel were similar to each other, and those of the cloud pixels were quite different. This phenomenon can be expressed in a more intuitive manner through the distance between $r$ and $g$.

To date, RBR has been widely used:

$$\text{RBR} = \frac{R}{B} = \frac{r}{b}, \tag{10}$$

The $r$ and $b$ of the clear sky pixel were close to each other, and the ratio of $r$ to $b$ was large. The $r$ of the cloud pixel was far from $b$, and the ratio of $r$ to $b$ was small. The same conclusion can be obtained by observing $r$ and $g$.

The NRBR is meant to represent the blueness of the sky and improves image contrast and robustness to noise:

$$\text{NRBR} = \frac{B-R}{B+R} = \frac{1-R/B}{1+R/B} = \frac{b-r}{b+r} = 1 - \frac{2r}{1-g} \approx 1 - 3r, \tag{11}$$

The NRBR is the normalization of RBR and is a monotonically decreasing function of the RBR. The RBR range was theoretically (0, 255) and the NRBR range was (−1, 1) (Figure 10a). However, the RBR range in the sky image was (0, 1), as was the NRBR range. Figure 10b shows the relationship between the two features. The NRBR was converted to the chromaticity coordinates. When the blue channel was not saturated, $g$ was approximately 0.34 and NRBR was approximately the function of $r$.

The S feature was designed to identify the saturation of the sky, and its theoretical basis is consistent with the features discussed above. According to the definition of chromaticity coordinates, S can be derived as the function of $r$, which is the same as the NRBR approximation function.

$$S = 1 - \frac{3}{R+G+B}\left[\min(R,G,B)\right] = 1 - \frac{3}{R/r} \times R = 1 - 3r. \tag{12}$$

To numerically understand the three features, the feature values on the red stripe in Figure 8a are plotted in Figure 9d. The trends of the three features were observed to be similar. The results of RBR and NRBR were fully compatible with the relationship shown in Figure 10b. Compared to RBR, although the nonlinear compression of NRBR reduced noise, it was prone to reduce the contrast between clouds and clear skies. The consistency of the chromaticity coordinate function of S with the approximation function of NRBR was verified. The analysis showed that RBR and NRBR measured the difference between $r$ and $b$, and S and NRBR were regarded as the measures of $r$. The measure of the difference between $r$ and $g$ can obtain similar results. Thus, the ARGD feature was proposed. ARGD was not only consistent with the theory, but also corrected the blue component saturation. The computational cost was minimized, and the adjustable weight was highly suitable for cloud detection considering the interference of sunlight.

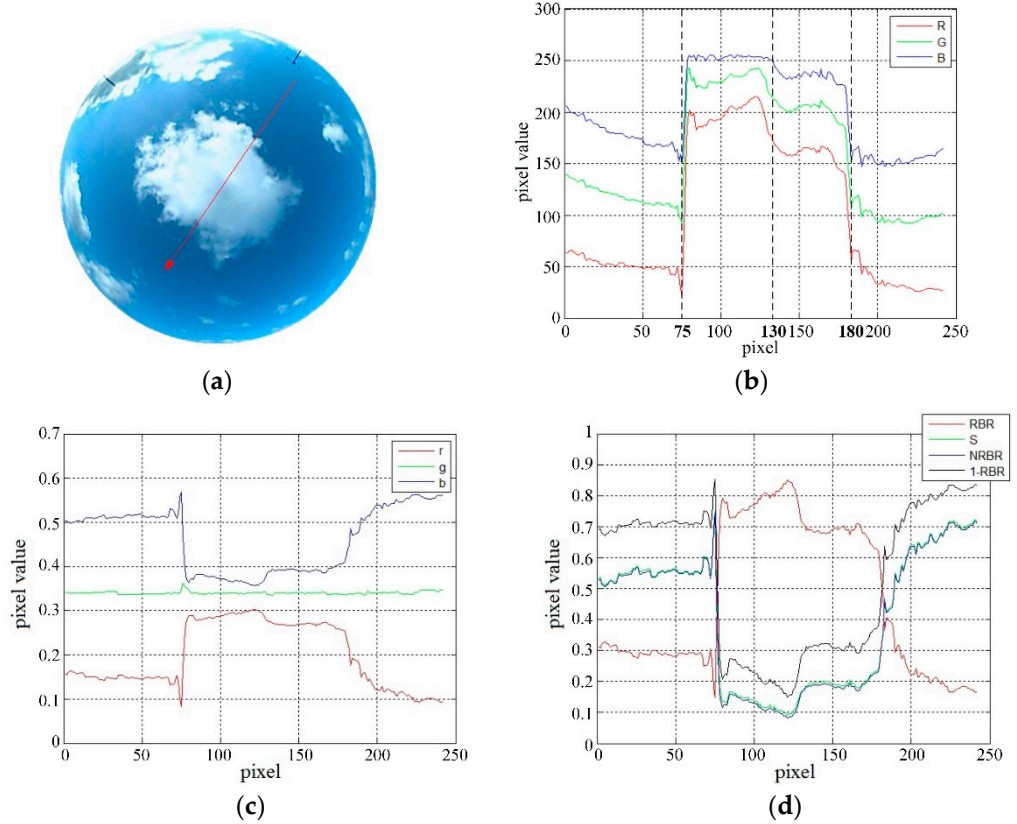

**Figure 9.** Discussion of the color channels and several features: (**a**) Original image with the red strip, (**b**) linear histogram of RGB channels, (**c**) linear histogram of the chromaticity coordinates, (**d**) relationship of the three features.

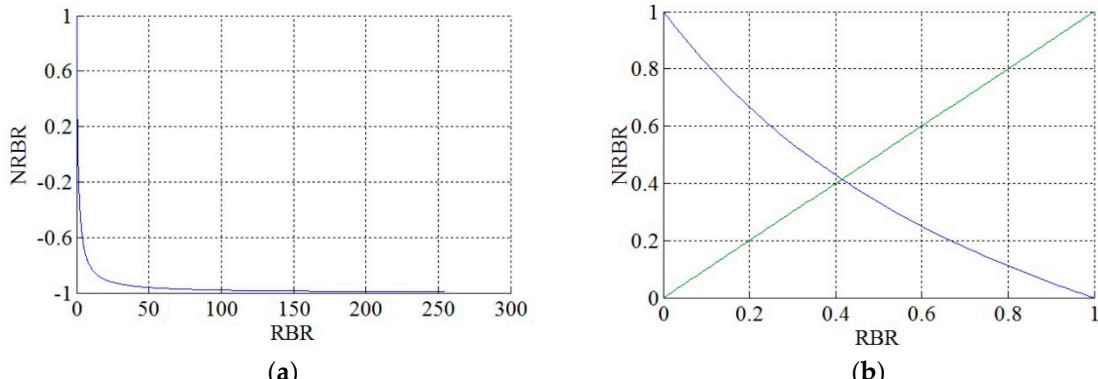

**Figure 10.** Relationships between NRBR and RBR: (**a**) Theoretical relationship; (**b**) actual relationship.

### 4.4. Comparison and Analysis

Our algorithm was compared to the algorithms using RBR with a fixed threshold, S with a fixed threshold, and HYTA, which flexibly uses fixed and minimum cross entropy (MCE) adaptive thresholds on the NRBR feature images.

The ARGD adopted the constant zero threshold, while the fixed threshold of other features were statistically estimated as described by [22]. A quarter of the image set was used to calculate the mean $\mu$ and the standard deviation $\delta$ of the clouds, and the threshold was finally set as $T = \mu \pm 3\delta$ based on the probability theory of Gaussian distribution, with $T_{RBR} = 0.36$, $T_S = 0.639$, and $T_{HYTA} = 0.452$. The clouds in various features had different properties. Therefore, the pixels for which the feature

values were smaller than the threshold were classified as clouds for S and HYTA, while the feature values of clouds must be greater than the threshold for RBR.

The numerical analysis of RBR, S, NRBR, and ARGD and image comparison of the corresponding algorithms are illustrated under different sky conditions (Figure 11). A linear histogram was plotted to analyze the performance of the pixels on the red line in the sky image under different features in detail. For better observation, ARGD was displayed in normalized form. Figure 11a shows the experimental results for a clear sky. It is observed from the linear histogram that cloud pixels were identified from 0 to 58 to RBR with the fixed threshold, and from 0 to 49 to S with the fixed threshold. Since the influence of sunlight caused the sky standard deviation to be large, the HYTA algorithm used the MCE adaptive threshold (0.529) on the NRBR, and clouds were identified from 0 to 67. The values of ARGD were negative for the entire red line, correctly identifying the clear-sky pixels at zero threshold. The misrecognition of other features in cloud pixels was considered to be caused by the interference of sunlight. Two parameters, namely SI and SD, were calculated to determine the sky state. The SI indicates that the brightness of the sun was high, so the sun appeared in the image. The SD indicates that the sky saturation was unevenly distributed, so the image was disturbed by sunlight. At this time, different layers had different weights. The performances of the four algorithms were also shown and it was observed that RBR detected more clouds than S, but fewer than HYTA, while ARGD reduced most of the interference of sunlight and showed excellent performance.

The image analyzed in Figure 11b contained both clouds and clear skies and suffered from interference by sunlight. For the RBR in the linear histogram, the cloud pixels were identified over the ranges of 0–64, 66–132, and 134–224, with the rest of the pixels identified as clear skies. Cloud pixels were identified over the ranges of 0–64, 66–124, and 134–217 in the S. NRBR used MCE adaptive thresholds of 0.6, 0–228, and 232–237 to be identified as clouds. The clouds were identified from 10–63 and 136–217 for ARGD, obtaining results that were consistent with the actual sky conditions. Due to the interference of sunlight, misrecognition was still present. The parameters of the sky state indicate that the sun appeared in the image and the image had sunlight interference, meaning the weight was determined. The detection results of the four algorithms coincide with the cloud distribution of each feature in the linear histogram. It can be seen that the ARGD result was particularly outstanding compared to other features when the image was interfered by sunlight. This indicates that ARGD successfully reduced solar interference.

The overcast sky is analyzed in Figure 11c. All four algorithms used the fixed threshold and the pixels on the entire red line were correctly identified as clouds. The SI indicates that the brightness of the sun was small, so the sun did not appear in the image. The SD indicates that the sky saturation was evenly distributed, so the image was not disturbed by sunlight. In this case, the weights of the layers were the same. The detection results of the algorithms correctly reflect the actual situation of the sky. Overall, ARGD performed well under three sky conditions and outperformed other algorithms when the image was interfered by sunlight.

A classic confusion matrix evaluation method [24] was used to assess the accuracy of detection. Four statistics can be clearly observed in the matrix shown in Table 4. True positive (TP) and true negative (TN) represent the correct detection for clouds and clear skies, respectively. False negative (FN) and false positive (FP) represent incorrect detection. Several classical evaluation indices, such as accuracy, recall, and precision, can be obtained using these four statistics. Accuracy, which is defined as the percentage of correct detection, is one of the most widely used evaluation indicators, and is used in many studies used as the sole criterion for the evaluation of the algorithm performance. In this work, recall can be interpreted as the proportion that is accurately detected in the clouds and precision is the fraction of the pixels predicted as clouds that are in fact cloud pixels. Although a higher accuracy indicates the better performance of the algorithm, it should be accompanied by the balanced values of the other criteria.

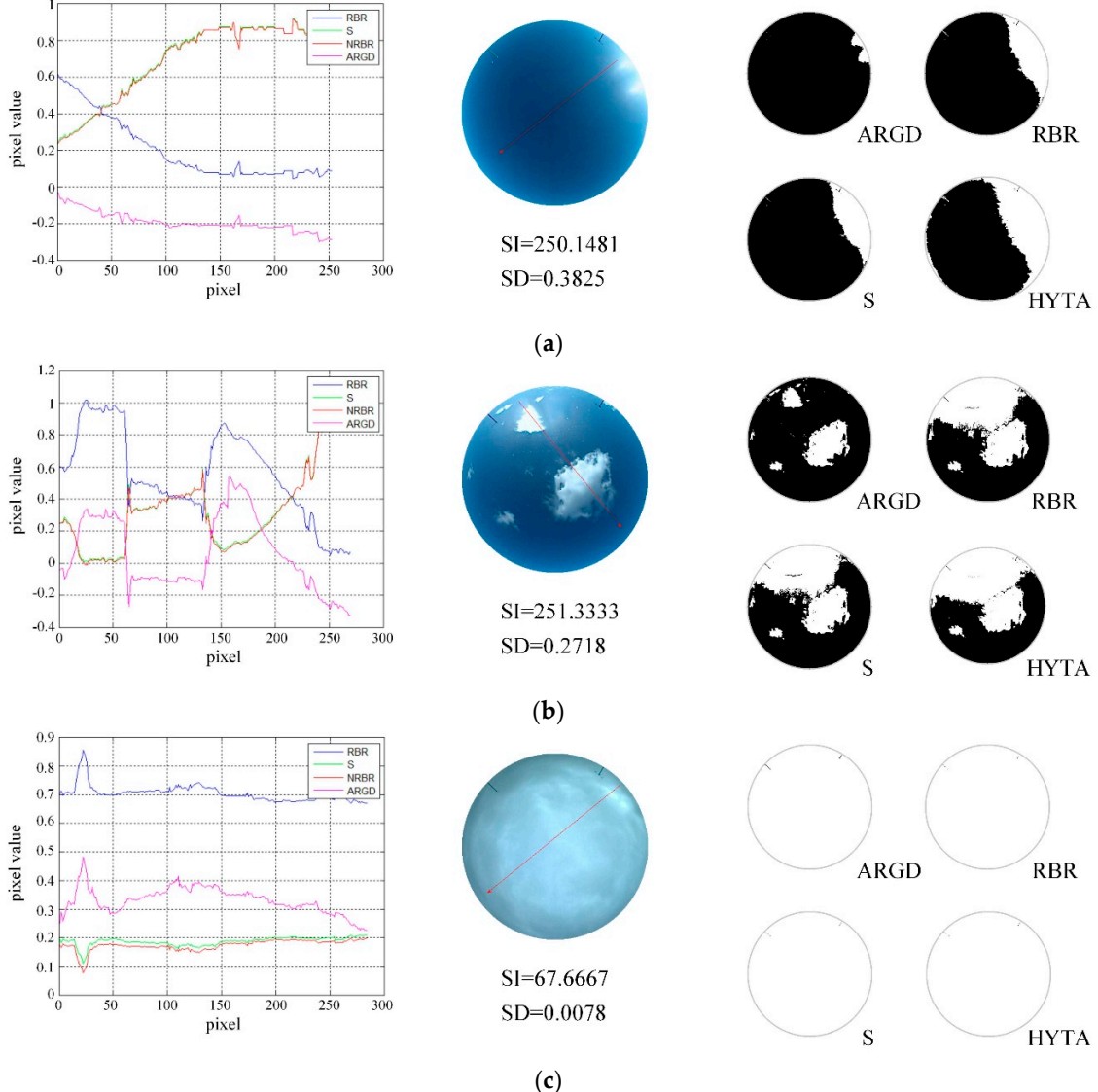

**Figure 11.** Numerical analysis of RBR, S, NRBR, and ARGD, and image comparison for the corresponding cloud recognition algorithms: (**a**) Numerical analysis and image comparison for the clear sky; (**b**) numerical analysis and image comparison for the cloudy sky; (**c**) numerical analysis and image comparison for the overcast sky.

**Table 4.** Confusion matrix and evaluation indicators.

| | | Actual | | |
|---|---|---|---|---|
| | | **Cloud** | **Clear** | |
| **Predicted** | **Cloud** | TP | FP | Precision $= \frac{TP}{TP+FP}$ |
| | **Clear** | FN | TN | |
| | | Recall $= \frac{TP}{TP+FN}$ | | Accuracy $= \frac{TP+TN}{TP+TN+FP+FN}$ |

The confusion matrix of ARGD for evaluating the performance of the algorithm is listed in Table 5 with the statistics expressed as percentages. The probability that clear skies were misclassified as clouds was as low as 1.34%, suggesting that the algorithm reduced the interference of sunlight. The case where the clouds were misclassified as clear skies was considered to be caused by the dark clouds

in the area interfered by sunlight. Overall, the correct classification of 98.02% is quite satisfactory, demonstrating the effectiveness of the algorithm.

**Table 5.** Confusion matrix for ARGD.

|  |  | Actual | |
|---|---|---|---|
|  |  | **Cloud** | **Clear** |
| **Predicted** | **Cloud** | 64.72 | 0.64 |
|  | **Clear** | 1.34 | 33.3 |

The evaluation results for the four algorithms are presented in the histogram (see Figure 12). The recall of RBR was 99.42%, which was the highest among the four features, indicating that the majority of actual clouds were identified. The precision indicates that 9.17% of the pixels should be clear skies but were falsely determined to be clouds, and the accuracy was 92.98%, ranking third among all features. The accuracy and precision of S were 94.04% and 92.29% respectively, and the recall was 6.97% higher than the precision. The three statistics for S showed the second highest values among the four algorithms.

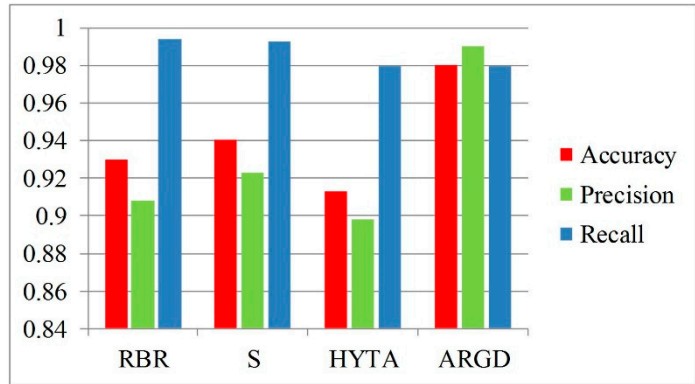

**Figure 12.** Evaluation results for several algorithms.

The accuracy of HYTA was the lowest at 91.29%. The recall of 97.97% is not low, but the precision of 89.78% is not good, so the tradeoff between these criteria was observed. Different threshold methods were employed to determine whether the image was unimodal or bimodal in this algorithm. The images of cumuliform and cirriform clouds segmented with adaptive thresholds were bimodal, and the images of stratiform clouds and clear sky with fixed thresholds for segmentation were unimodal. Different from the work of HYTA, the set of images in this paper was not image patches containing only a single cloud type. Clouds of different types inevitably appeared in an image to be processed, so the detection accuracy was decreased. The sky in a small area was uniform to the color change, but the entire clear sky was interfered by the uneven sunlight, so the unimodal result was detected. This is one of the reasons why HYTA performed poorly in this case. In addition to the interference of sunlight, the presence of multiple peaks also gave rise to the error of the MCE threshold.

Compared to other algorithms, ARGD had the highest accuracy of 98.02%, and moreover, the recall and precision were the most balanced. Although the recall rate of 99.02% was the lowest, it was not much different from the recall values of the other algorithms. Overall, ARGD obtained outstanding detection results. The outstanding performance was due to the weakening of the effect of sunlight, demonstrating that interference by sunlight was the difficulty that must be overcome to obtain improved accuracy. In summary, ARGD reduced the interference of sunlight and achieved better performance. Therefore, it is the most promising approach.

## 5. Conclusions

We proposed a cloud detection algorithm for reducing the interference of sunlight in ground-based sky images. The proposed solar location method locates the pixel coordinates of the sun using the coordinate transformation. The coordinate transformation formula is obtained by fitting the polynomial surface with manually labeled pixel coordinates and horizontal coordinates. This overcomes the drawbacks of the lack of device information and calibration data. The ARGD feature using red and green channels takes into account the interference of sunlight in the image, which is reflected in the red weight determined by the layering region division and the sky state. The layering region division classifies the degree of sunlight interference in the image by tracking the position of the sun. The sky state discriminates whether there is sunlight interference in the image according to the SI and SD. After the image is calculated by the feature, the zero threshold is used for segmentation. Our experiments demonstrate that ARGD effectively attenuates the interference of sunlight and outperforms other algorithms.

**Author Contributions:** Conceptualization, X.L. and Z.L.; Methodology, X.L. and Z.L.; Resources, Z.X.; Software, X.L.; Validation, X. and Q.Z.; Writing–review & editing, X.L. and Q.Z.

**Funding:** This research was funded by the National Natural Science Foundation of China, grant number 51677123.

**Acknowledgments:** Authors would like to thank Atmospheric Radiation Measurement (ARM) Program sponsored by the U.S. Department of Energy, Office of Science, Office of Biological and Environmental Research, Climate and Environmental Sciences Division for image data support.

**Conflicts of Interest:** The authors declare no conflict of interest.

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
