# Peer review of "A Cloud Detection Algorithm with Reduction of Sunlight Interference in Ground-Based Sky Images"

_atmosphere, doi:10.3390/atmos10110640_

Round 1

Reviewer 1 Report

This study develops a new algorithm for detecting boundary layer clouds with ground-based sky images. The major merit of this method is that it is more effective in reducing sunlight interference, a known problem in ground-based cloud detection. Overall, the authors did a good job in describing the detail of the method and analyzing the results. The only weakness, however, is that it is somewhat too technical with little discussion on scientific implication to the atmospheric science community. After all, people who are interested in cloud detection and use TSI retrievals are mostly atmospheric scientists. A very important question this study should address is how the improved algorithm impacts the relevant cloud research. In other words, how important this study is?  If this issue can be effective addressed, I recommend accepting this manuscript. The detail comments are as follows.

Major comments:

It is well known that the clouds over the ENS have distinctive diurnal and seasonal cycles (Robert Wood's BAMS paper). Researchers use the TSI cloud retrievals to understand these cloud behaviors, which have many important implications for climate change research. The authors should at least discuss the implications of the method under such context. It would be great if the author can use this new method to show the diurnal and seasonal cycle of cloud coverage, from which the authors can extract much more useful information (such as the robustness of the new method and new insights on cloud climatology over Azores). Also, some introduction to the importance of clouds should be added in the introduction section. Anyway, this journal is not a technique-centered journal, some scientific context is very needed. 

Minor comments:

Lines 29~30: redundant sentence

Figures 1, 9, 10, 11: the fonts of x and y axes are too small to read. Make them larger. 

Author Response

1. Response to comment: The only weakness is that it is somewhat too technical with little discussion on scientific implication to the atmospheric science community.

Response: we are very sorry for our negligence of the discussion on    scientific implication to the atmospheric science community. A discussion on scientific implication to the atmospheric science community was added in Lines 22-24.

2. Response to comment: A very important question this study should address is how the improved algorithm impacts the relevant cloud research. In other words, how important this study is?

Response: the importance of this study was added in Lines 27-29.

3. Response to comment: It is well known that the clouds over the ENS have distinctive diurnal and seasonal cycles (Robert Wood's BAMS paper). Researchers use the TSI cloud retrievals to understand these cloud behaviors, which have many important implications for climate change research. The authors should at least discuss the implications of the method under such context. It would be great if the author can use this new method to show the diurnal and seasonal cycle of cloud coverage, from which the authors can extract much more useful information.

Response: The implications of the method under such context that the clouds over the ENS have distinctive diurnal and seasonal cycles was added in Lines 24-26, 27-29. The implications of the method for other cases was added in Lines 26-27 because the goal of our paper is to apply to multiple-fields cloud detection. Limited by resources and time, we only used this new method to show the diurnal of cloud coverage in August as shown in the following image.

The periodicity of cloud coverage is a part of our subsequent researches, so more detailed research and analysis will be developed in the future work.

4. Response to comment: Some introduction to the importance of clouds should be added in the introduction section.

Response: some introduction to the importance of clouds was added in Lines 22-24.

5. Response to comment: Lines 29~30: redundant sentence

Response: the redundant sentence was deleted in Lines 29~30

6.Response to comment: Figures 1, 9, 10, 11: the fonts of x and y axes are too small to read. Make them larger.

Response: We have adjusted the font size of axes in the figures according to the reviewer’s comments.

Special thanks to you for your good comments.

Reviewer 2 Report

Journal ATMOSPHERE MDPI 610077                                        September 24th 2019

Title: “A Cloud Detection Algorithm with Reduction of Sunlight Interference in Ground-based Sky Images”

Authors: Xin Li 1, Zhiying Lu 1, *, Qingxia Zhou 1 and Zhengyang Xu 1 4

1 School of Electrical and Information Engineering, Tianjin University, Tianjin 300072, China  

Abstract: Cloud detection for ground-based sky images has attracted much attention in  cloud-related fields. In this paper, we propose a cloud detection algorithm that reduces the sunlight interference in the image. The solar location method is introduced to track the sun in the image  used for feature calculation, which is suitable for the case where the camera cannot be calibrated.  Following this, the adjustable red green difference (ARGD) feature using red and green channels is  proposed. The red weight in the feature is determined by the layering region division that classifies  the degree of sunlight interference in the image and the sky state that discriminates whether there  is sunlight interference in the image. Finally, a fixed zero threshold is applied to feature images in  order to obtain the cloud detection results. Experimental results show that the proposed algorithm  performs better than the other algorithms and can effectively reduce the sunlight interference.  

Keywords: clouds detection; sunlight interference; solar location; ground-based images; image  processing.

Recommendations:

1-Figure 1: The number size scales along the OX and OY axis should be increased. Thank you.

2-Figure 2a:  Letter size on Fig 2 (a) is small for reading easily. Please change it.

3- Figure 5: The units of SZA and SAA should be included. Please also include the units of row and column. What magnitude are they? And what units they have??.More details in relation to the magnitudes should be  included.

4-Figure 9:  Axis divisions and axis titles are writing in very small sizes. Please correct them.

5- Figure 10: Axis titles are writing very small. Please correct them.

6- Similar to Figure 10 and in Figure 11. Please correct it.

7- Figure 12 format should be similar to previous Figures.

8- In equations 7 and 8  what are the units of RMSE???

9-Table 1 and Table 2 RMSE has units. In affirmative case, write them.

10-It is not clear the magnitude of the data series that the manuscript used.

11-The characteristics of the data series used have to be clearly explained. For doing that, it would be better to read the publication that is described down:

12- The manuscript shows a cloud detection method for solar location in ground-based sky images. The method used the coordinate transformation. The manuscript is clearly written but it needs explain with more detail some of the variables and magnitudes used in order to increase the results quality.

Author Response

1. Response to comment: Figure 1: The number size scales along the OX and OY axis should be increased.

Response: The number size scales along the OX and OY axis have been increased in Figure 1.

2. Response to comment: Figure 2a: Letter size on Fig 2 (a) is small for reading easily. Please change it.

Response: Letter size on Figure 2a was increased.

3. Response to comment: Figure 5: The units of SZA and SAA should be included. Please also include the units of row and column. What magnitude are they? And what units they have? More details in relation to the magnitudes should be included.

Response: The units of SZA, SAA, row and column are added in Figure 5. More details about SZA and SAA are added in Lines 167-170. In order to improve the quality of the presentation, some corrections about SZA and SAA are made in Lines 163-166. More details about row and column are added in Lines 180-181.

4. Response to comment: Figure 9: Axis divisions and axis titles are writing in very small sizes.

Response: The size of axis divisions and axis titles was increased in Figure 9.

5. Response to comment: Figure 10: Axis titles are writing very small.

Response: The size of axis titles was increased in Figure 10.

6. Response to comment: Similar to Figure 10 and in Figure 11.

Response: The size of axis titles was increased in Figure 11.

7. Response to comment: Figure 12 format should be similar to previous Figures.

Response: Letter size and the size of axis divisions and axis titles were increased in Figure 12.

8. Response to comment: In equations 7 and 8, what are the units of RMSE?

Response: In this paper, the RMSE describes the fitting error of pixel coordinates row and column and the units of RMSE are pixel. A description of the units was added in Lines 303-305.

9. Response to comment: Table 1 and Table 2 RMSE has units. In affirmative case, write them.

Response: The units of RMSE were added in Table 1 and Table 2.

10. Response to comment: It is not clear the magnitude of the data series that the manuscript used.

Response: the magnitude of the data series was added in Line 107. Lines 115-118 are also related to the data series, but they were not placed in a paragraph considering the structure of the article.

11. Response to comment: The characteristics of the data series used have to be clearly explained.

Response: The characteristics of the data series used were added in Lines 107-108. It's a pity that the review opinion does not show the publication you recommended, so we added it according to other understanding and hope that the content will meet with approval.

12. Response to comment: The manuscript is clearly written but it needs explain with more detail some of the variables and magnitudes used in order to increase the results quality.

Response: An instruction for units and ranges of SD was added in Lines 230-231.

An instruction for units and ranges of SD was added in Lines 241-242.

A description of the units about the Adjusted R-square was added in Lines 303-305 and the units of Adjusted R-square were added in Table 1 and Table 2.

Thank you very much for your comments and suggestions.

Round 2

Reviewer 1 Report

In the revised version, the authors have well addressed my comments. I have no problem with it being accepted.